# Exploring confidence development in interprofessional teams: A pre-post analysis of a health and social care education module

Sharron Blumenthal[1], Sivaramkumar Shanmugam[1*], Jamie McDermott[2],
John Locke[3], Tamsin Fitzgerald[1], Christopher Duncan[1], Kareena McAloney-Kocaman[4],
Lindsey Burns[4]

**1** Department of Physiotherapy and Paramedicine, School of Health and Life Sciences, Glasgow Caledonian University, Glasgow, United Kingdom, **2** School of Health and Life Sciences, Glasgow Caledonian University, Glasgow, United Kingdom, **3** Department of Podiatry and Radiography, School of Health and Life Sciences, Glasgow Caledonian University, Glasgow, United Kingdom, **4** Department of Psychology, School of Health and Life Sciences, Glasgow Caledonian University, Glasgow, United Kingdom

☉ These authors contributed equally to this work.
* Sivaram.shanmugam@gcu.ac.uk

## Abstract

### Aim

Confidence can be defined as a strong belief, firm trust, or sure expectation in relation to achieving an outcome. The study aimed to examine change in confidence to collaborate in teams in undergraduate health and social care students undertaking a mandatory 6-week IPE module using the Team Observed Professional Encounter (TOSPE).

### Method

A pre-test post-test study was undertaken. Confidence was measured using the validated Interprofessional Education Academic Behavioural Confidence Scale. Independent t-tests and Analysis of Variance were used to establish differences between groups at the commencement of the study. To compare pre and post confidence scores paired t-tests were used for normally distributed data, and Wilcoxon signed rank test were undertaken for non-normally distributed data. Cohen's d and Pearson r were produced as measures of effect size. A mixed design Analysis of Variance was conducted to examine the influence of categorical variables on changes in confidence scores.

### Results

Data were included from 80 matched pairs. Pre-test self-perceived scores for Total Confidence and the subscales for Interprofessional Team Working, Interprofessional

**Data availability statement:** Yes - All data are in the manuscript and/or supporting information files.

**Funding:** The author(s) received no specific funding for this work.

**Competing interests:** The authors have declared that no competing interests exist.

Communication and Behaviours Underpinning Collaboration increased significantly (p =<.001) post module completion. Changes in scores demonstrated large effect sizes for total confidence (d = .888), Interprofessional Team Working (d = .872), Interprofessional Communication (d = .945) and a medium effect size for Behaviours Underpinning Collaboration (r = .534). Time was found to be the only categorical variable that had a significant effect on confidence (p < 0.05).

## Conclusion

Post-intervention student confidence increased significantly for total confidence and all subscales of the IPE ABC scale. When considering between subject effects only time was found to demonstrate significant results indicating an association between the IPE intervention and increased self-perceived confidence. However, it should be noted that other factors such as small subgroup sample size, social desirability response bias and potential overconfidence bias may impact the results observed, so caution in interpretation of the results considering these limitations are advised. This study adds to the body of literature that suggests IPE interventions impact positively on behaviours that underpin collaborative practice.

## Introduction

Interprofessional collaborative practice (IPC) is a critical strategy purported to address the global health workforce crisis [1,2]. IPC is defined as when professionals from different health or social care backgrounds work together to provide services [3]. Effective IPC enhances teamwork, which is vital for effective communication and co-ordination of care. [4]. IPC enhances the delivery of person-centred care along with improved patient outcomes [5,6]. Patient benefits of IPC include better patient education, self-management skills, care transitions, satisfaction and reduced hospital stays [7]. Professional and organisational outcomes also benefit from IPC, with reported increases in job satisfaction and performance, efficiency, perceived autonomy, and reduced burnout [4–7].

For IPC to be effective, practitioners must be trained to work in interprofessional teams, and a key way to achieve this learning is through interprofessional education (IPE) [1,8,9]. The Centre for the Advancement in Interprofessional Education (CAIPE) defines IPE as "*occasions when members or students of two or more professions learn with, from and about each other to improve collaboration and the quality of care and services*" [10]. In the UK IPE is a mandated component of health and social care education, supported by professional and regulatory bodies such as the Health Care Professions Council [11], the Nursing and Midwifery Council [12], the General Medical Council [13], the General Dental Council [14], and the Scottish Social Services Council [15]. Despite widespread endorsement of IPE, few studies have evaluated validated tools in pre-registration academic settings using robust pre-post designs.

## Background

The Glasgow Caledonian University (GCU) IPE framework spans over years 1–3 of undergraduate programmes following the University of British Columbia (UBC) model for IPE, which includes three phases: exposure, immersion, and mastery [16]. Over 4000 students from fifteen different disciplines engage in this framework annually (Fig 1). These disciplines include Nursing (Adult, Child, Mental Health, and Learning Disability branches), Allied Health Professions (Diagnostic Imaging, Human Nutrition and Dietetics, Occupational Therapy, Orthoptics, Physiotherapy, Podiatry, and Radiotherapy and Oncology), Dental (Oral Health Science) and Social Work programmes. Prosthetics and Orthotics students attend from Strathclyde University (SU) for the year one module.

Confidence, defined as a strong belief or expectation in achieving an outcome [17], is a key factor influencing students' ability to participate effectively in interprofessional learning and collaborative practice [18]. Oxlad et al [19] found that confidence impacts students' ability to actively contribute to IPE, emphasizing the need for educators to create safe learning environments that foster confidence. Confidence is underpinned by self-efficacy, and expectancy value theory [20]. Students' self-efficacy is related to motivation and achievement and influences student engagement, persistence, and effort in academic learning activities [21]. Expectancy-value theory suggests that study motivation is underpinned by student beliefs about not only how well they will do on an activity, but also the extent to which that activity is valued [20]. Hermansen-Kobulnicky et al. suggest that measuring student confidence is integral to evaluating student perceived ability regarding interprofessional collaborative practice [22].

A small number of studies have explored the impact of IPE on student confidence in relation to interprofessional collaboration and engagement with interprofessional education. Street et al. identified that nursing students lacked confidence to engage within IPE activities with medical students due to their misguided perceptions of their own knowledge when compared to medical students [23]. Forbes et al. noted that on transition to practice nurses' confidence impacted on their communication with physician's [24]. Furthermore, the same study identified that as nurses self-perceived confidence grew in their new roles so did their communication with medical staff [24]. Oxlad et al explored psychology and dental student confidence and its impact on IPE through qualitative analysis of an IPE intervention [19]. The authors identified that confidence impacted on student ability to communicate in interprofessional groups. Furthermore, psychology students lacked confidence in themselves as they felt dental students would know more than them. However, the study also

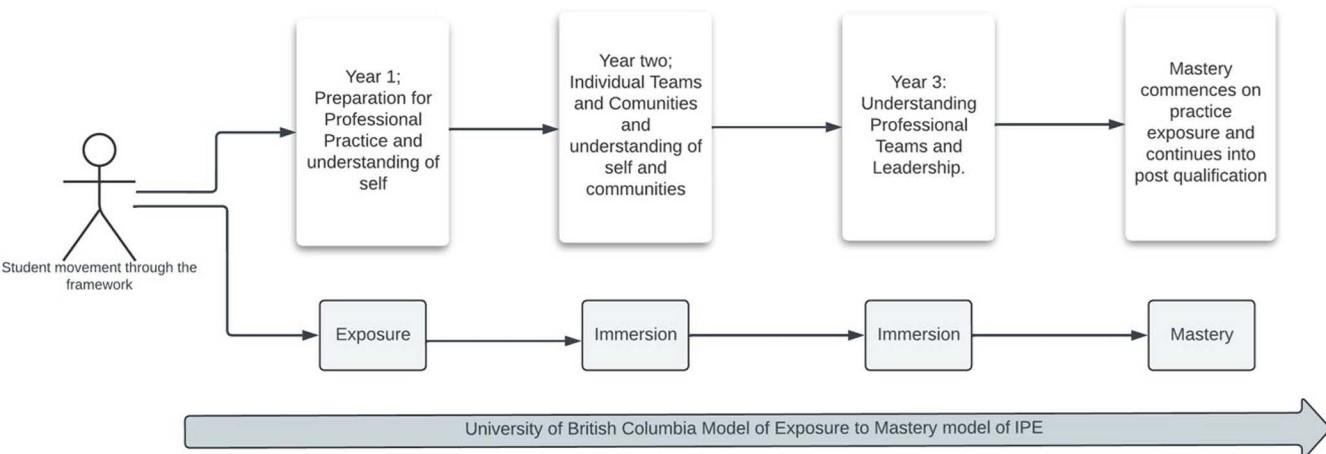

**Fig 1. GCU Undergraduate Pre-registration Interprofessional Education Framework based on the University of British Columbia Model of IPE.**

identified that through the IPE interaction students noted their confidence increased and they felt further IPE exposure was essential to the development of confidence in interprofessional interactions.

Hermansen-Kobulnicky et al. undertook a retrospective pre-post-test study of dietetics, nursing, social work, pharmacy, speech and language and hearing science student confidence and identified increased confidence levels in students post the IPE intervention in the academic environment [22]. However, a limitation of this study was that Hermansen-Kobulnicky developed a non-validated scale for confidence based on the Interprofessional Education Collaborative sub-competencies [22].

McLaren, Muston, and Page assessed self-perceived confidence and attitudes toward interprofessional learning pre and post a simulated ward session in pre-registration nursing, physiotherapy, and medical students [19]. The results demonstrated that student confidence increased post the intervention and there was a significant change in student attitudes towards interprofessional learning using the Readiness for Interprofessional Learning Scale (RIPLS) indicating that increased confidence impacts positively on student attitudes to interprofessional learning. Thus, it is reasonable to postulate that IPE interventions can increase student confidence and that an increase in confidence can impact on attitudes towards interprofessional learning however the tools used to measure confidence in both studies were not valid and reliable. Further studies are needed to assess confidence using validated scales.

The Interprofessional Education Academic Behavioural Confidence (IPE ABC) scale is a self-report questionnaire that measures student confidence levels to undertake IPE related activities within an academic environment. Blumenthal et al. developed the questionnaire to measure health and social care students' confidence to engage in IPE within the academic setting [25]. Fifteen different professional groups of pre-registration nursing, allied health, and Social Work students (n = 565) participated in the development of the questionnaire. Exploratory factor analysis identified three factors: 1/ interprofessional teamwork, 2/ behaviours underpinning collaboration, and 3/ interprofessional communication collectively accounting for 38.2% of the variance. Internal consistency of the overall scale (Cronbach's $\alpha$ = .93) was very good with subscales demonstrating very good internal consistency, 1 ($\alpha$ = .89), or respectable consistency 2 ($\alpha$ = .78) and 3 ($\alpha$ = .79). The questionnaire was deemed to be valid and reliable [25]. Table 2 shows the items within the IPE-ABC scale and which subscale each item belongs to.

Despite the recognised importance of IPC and IPE, there is limited empirical evidence on the impact of academic IPE modules on student confidence levels in undertaking IPC activities. Previous studies have used non-validated scales leaving a gap in understanding the effects of IPE on student confidence.

The current study aims to address this gap by evaluating changes in student confidence after completing an academic based IPE module using the validated IPE ABC scale. This study will provide insights into the effectiveness of IPE modules in enhancing student confidence and preparedness for IPC.

## Methods

This study was a single cohort pre-test post-test study involving year three undergraduate students enrolled in the mandatory UTPL module during the second trimester commencing February 2024. The study was carried out at GCU in Scotland. This study was reported in accordance with the STROBE (Strengthening the Reporting of Observational Studies in Epidemiology) guidelines for cohort studies [26].

### Participants

A convenience sample of four hundred and ninety-four year three students from Nursing BSc and BSc (Hons) programmes (Adult, Child, Learning Disability and Mental Health branches), and BSc (Hons) Allied Health programmes, (Diagnostic Imaging, Occupational Therapy, Podiatry and Radiotherapy and Oncology) were invited to participate.

For students to take part in the study they needed to attend the module sessions. Participation in the study was voluntary, and students could withdraw from the survey at any time. Students were also informed that completion or

non-completion of the survey, would not affect their course progression, marks, or relationship with the University. Ethical approval was obtained from the University Ethics Committee (HLS/PSWAHS/23/107). The sample size was determined by the number of students enrolled in the module during the study period, as this study formed part of the routine module evaluation. As such, no a priori power calculation was performed. However, post hoc consideration of the large observed effect sizes suggest that the final matched sample was sufficient to detect meaningful pre-post changes at group level.

## Recruitment and consent

Recruitment materials, including a video explaining the research, a participant information sheet, and online links to the questionnaire were provided to students during week 1 (pre-test) and weeks 5, 6, and 7 (post-test), via the university virtual learning environment; GCU-Learn. Between February 2024 and March 2024, participants completed the pre and post online questionnaire. Participants indicated their consent by choosing 'yes' on the consent question at the start of the online questionnaire. If they selected 'No' they were exited from the online questionnaire. Before beginning, participants received comprehensive information about the research through both the questionnaire introduction and a detailed participant information sheet. This documentation explained the research objectives, guaranteed participant anonymity and data confidentiality, and stressed the voluntary nature of participation. The opening questionnaire question reinforced that proceeding with the questionnaire constituted agreement for their responses to be used in research. Participants retained the right to withdraw from the study at any point without consequences. To maintain data security, only authorised research team members could access the questionnaire responses, which were stored securely on password-protected systems. To prevent potential bias, the researchers analysing the data were separate from those involved in teaching or grading the module.

## The Intervention

The mandatory UTPL module runs over six-weeks and learning outcomes relate to interprofessional leadership, interprofessional identity and culture, human factors, and known competencies relating to interprofessional teamwork and collaborative practice, based on the frameworks from Canadian Interprofessional Health Collaborative (CIHC) [27] and Interprofessional Education Collaborative (IPEC) [28]. The module uses a flipped classroom model, where students engage in asynchronous learning activities covering various concepts, which are reinforced, and further explored within in a two-hour tutorial. The first hour of the tutorial contextualises theoretical constructs such as leadership in an interprofessional context. The second hour in weeks 2–5, focuses on the development of collaborative competence using the Team Observed Structured Professional Encounter (TOSPE) method [29]. Assessment for the module consisted of an individual 2000 word written coursework relating to patient safety and interprofessional leadership which accounted for 60% of the mark and a summative TOSPE which accounted for 40% of the mark.

Within the TOSPE students work in groups of 6–7 and are presented with a case study on a weekly basis which they work with as an interprofessional team implementing a team meeting to develop a collaborative management plan. Meetings are structured using the NHS Education for Scotland (NES) seven step meeting process [30]. The TOSPE evaluates the team's proficiency across the following domains: interprofessional communication and collaboration; management of conflict; interprofessional team functioning; role and responsibilities (professional and interprofessional); and human factors. Students undertook four formative TOSPE sessions, receiving feedback on these on a weekly basis, via a simulation style pre-brief and debrief approach. In week 6 students underwent a summative TOSPE assessment. Module content and structure is presented in the Table 1.

## Outcome measure

The IPE ABC scale [25] was used as a pre and post-test measure. The scale contains 38 items that were graded on a 4-point Likert scale with 1 being "not very confident" and 4 being "very confident". There was a fifth option for students to state if they felt unsure about their level of confidence. The Likert scale structure was chosen in

**Table 1. Module content and structure UTPL module.**

| Week | Pre-session asynchronous content | Classroom activity (2 hours) |
|---|---|---|
| 1 | Leadership theories | Exploration of the theories of leadership<br>The TOSPE approach<br>Pre-test questionnaire and recruitment information provided to students |
| 2 | Human Factors and patient safety | Human Factors and patient safety in an interprofessional context<br>Formative TOSPE |
| 3 | Professional identity | Professional identity in an interprofessional context<br>Formative TOSPE |
| 4 | Cultural competence | Professional culture in an interprofessional context<br>Formative TOSPE |
| 5 | Leadership | Leadership in an interprofessional context<br>Formative TOSPE<br>Post-test questionnaire provided to students |
| 6 | Pre for TOSPE assessment. | Summative TOSPE Examination<br>Reminder Post-test questionnaire provided to students |
| 7 | | Reminder Post-test questionnaire provided to students |

the original scale development study to reduce central tendency bias and to encourage respondents to select either a more positive or negative confidence rating rather than a neutral option, thus enhancing response discrimination [25]. The IPE ABC scale measures total confidence (TC) and is underpinned by three sub-scales: Interprofessional Teamwork (IPW), Behaviours Underpinning Collaboration (BUC), and Interprofessional Communication (IC) (Table 2).

## Scoring criteria

The scoring criteria established during the development of the IPE-ABC scale by Blumenthal et al. [25] were utilised for this study. Respondents could score a maximum of 152 and a minimum of 0 for total confidence in IPE activities. The Interprofessional Teamwork (ITW) subscale contained 20 items (max score 80), while the Behaviours Underpinning Collaboration (BUC) and Interprofessional Communication (IC) subscales each contained 9 items (max score 36 each). The scoring criteria and associated confidence levels are presented in Table 3.

## Data collection

Data was collected via the use of an online version of the IPE ABC scale set up on Microsoft Forms. The form was set to ensure anonymity and confidentiality. Survey completion had consent questions within the first page confirming student's willingness to take part in the survey and for their results to be used for research purposes. To ensure anonymity participants were asked to generate an alias code known only to them. Following data collection pre and post-test surveys were matched using the alias codes.

## Data analysis

**Data entry & cleaning.** all survey responses were exported from Microsoft Forms into SPSS [version 28] statistical software package for analysis. Random checks were undertaken to ensure accurate data input. No errors in data entry were identified. Data cleaning involved checking for missing values, outliers, and inconsistencies. 352 responses were obtained (223 pre-test questionnaires and 129 post-test questionnaires). Two questionnaires were excluded from any analysis (one due to missing profession data and one due to the participant not consenting to data being used for research purposes).

**Table 2. Subscales within the IPE ABC scale; adapted from Blumenthal et al. [25].**

| Items within the IPE-ABC Scale | Subscale |
|---|---|
| 1 Rely on students from a *different* professional group to undertake allocated team tasks. | Teamwork |
| 2 Engage in debate with students from a *different* professional background. | Teamwork |
| 3 Reflect on your individual performance in relation to team working | Teamwork |
| 4 Listen to the views of students from *different* professional backgrounds. | Behaviours underpinning collaboration |
| 5 Demonstrate respect for the skills of fellow students from *different* professional backgrounds. | Behaviours underpinning collaboration |
| *6. Develop rapport with fellow students from different professional groups.* | *Interprofessional Communication* |
| 7 Perform effectively in an interprofessional team as a team member when undertaking an academic task. | *Interprofessional Communication* |
| 8 Talk openly if things are not working with fellow students | *Interprofessional Communication* |
| 9 Work collaboratively to overcome barriers to teamwork. | *Interprofessional Communication* |
| 10 Feedback to fellow students on their performance | *Interprofessional Communication* |
| 11 Work collaboratively to develop clear goals | *Interprofessional Communication* |
| 12 Demonstrate you value the views of fellow students, even if they differ from your own. | Behaviours underpinning collaboration |
| 13 Resolve conflict effectively when working with fellow students, even when conflict becomes personal. | *Interprofessional Communication* |
| 14 Verbally confront fellow students who express negative judgements against other professional groups. | *Interprofessional Communication* |
| 15 Identify issues that have led to team breakdown. | *Interprofessional Communication* |
| 16 Discuss your understanding of your professional roles and responsibilities with students from *different* professional groups. | Teamwork |
| 17 Communicate effectively with other students. | Teamwork |
| 18 Establish clear team goals. | Teamwork |
| 19 Use common terminology when communicating with students from *different* professional backgrounds. | Teamwork |
| 20 Assist interprofessional team members at times of difficulty. | Teamwork |
| 21 Feedback to fellow students in a way that promotes team development when undertaking interprofessional activities. | Teamwork |
| 22 Engage fully in activities undertaken as part of interprofessional teamwork. | Teamwork |
| 23 Demonstrate respect for the roles and responsibilities of other professions in relation to your own. | Behaviours underpinning collaboration |
| 24 Reflect on experiences of working in interprofessional teams | Teamwork |
| 25. Recognise interprofessional learning activities with equal importance to your discipline specific learning activities. | Teamwork |
| 26 Recognise students from *different* professional backgrounds as equal to students from your own professional background. | Behaviours underpinning collaboration |
| 27 Agree ground rules for accepted team behaviours and methods of communication | Behaviours underpinning collaboration |
| *28 Engage in activities that explore the roles and responsibilities of other interprofessional team members.* | Behaviours underpinning collaboration |
| *29 Work actively to develop an understanding of how different professional groups could work together.* | Behaviours underpinning collaboration |
| 30 Recognise if stereotypical judgments are being made in relation to other professional groups. | Teamwork |
| 31 Undertake fully your agreed roles when working with fellow students from *different* professional backgrounds. | Teamwork |
| 32. Recognise and observe the limitations of your own professional role. | Teamwork |
| 33. Reflect on team performance. | Teamwork |
| 34. Perform effectively in an interprofessional team as a leader when undertaking an academic task. | Teamwork |

*(Continued)*

**Table 2.** (Continued)

| Items within the IPE-ABC Scale | Subscale |
|---|---|
| 35. Recognise the competence of other professions in relation to your own. | Teamwork |
| 36. Demonstrate you respect the contributions of fellow students | Behaviours underpinning collaboration |
| 37. Feedback to fellow students in a way that promotes positive team interactions when undertaking interprofessional activities. | Teamwork |
| 38. Utilise interprofessional experience to guide your future learning. | Teamwork |

**Missing data.** Only 43 cells within SPSS were incomplete. Due to the small number of cells that had no coded input, imputation was undertaken as a statistical technique to deal with missing data. A mean replacement method was undertaken due to the random nature of the missing cells [31]. This method was selected because it is appropriate when the missing data is assumed to be missing or random, and the proportion of missing data is low [32]. This approach is consistent with standard practices and helps to maintain the integrity of the dataset without significantly distorting the statistical analysis. While mean imputation was used to address a small proportion of missing responses, this approach assumes data were missing at random and may slightly reduce the natural variability in the dataset, however, given the low parentages of missing cells, any impact on the results is expected to be minimal.

**Descriptive stats.** For initial analysis descriptive stats were calculated for all demographic variables [i.e., gender, age, profession] and baseline confidence scores.

**Questionnaire analysis confidence.** Complete IPE-ABC scale mean scores were reviewed to identify the mean level of Total Confidence for the questionnaire. Mean subscale scores were reviewed to identify the mean level of confidence within each subscale. The normality of data was analysed using an assessment of skewness and kurtosis measurement [33]. We performed the Kolmogorov-Smirnov test and Shapiro -Wilk test which are commonly used for testing normality, test values of $< 0.05$ were used to determine non-normally distributed data [33].

Pre-test scores were normally distributed, and independent t tests and Analysis of Variance (ANOVA) were used to establish if there were any differences between groups at the commencement of the study. Statistical significance was set at $p = 0.05$

Paired t-tests were used for normally distributed data, to compare pre and post confidence scores. For non-normally distributed data, Wilcoxon signed rank test was used as a non-parametric test to assess the difference in pre and post confidence scores. Statistical significance was set at $p = 0.05$. Cohen's d and Pearson r were produced as measures of effect size. Values of 0.2, 0.5 and 0.8 indicated small, medium, and large effect sizes for both measures [33,34].

A mixed design Anova was conducted to examine the influence of categorical variables (time gender, age, and discipline) on changes in confidence scores. This analysis aimed to identify any significant interactions between these variables and confidence scores [32]. Statistical significance was set at $p = 0.05$.

# Results

## Response rates

A total of 494 students were eligible to participate in the research. Out of these, 223 students completed the pre-course IPE ABC Scale, yielding a response rate of 45%, while 128 students (response rate of 26%) completed the post-course IPE ABC scale, resulting in 80 matched pairs. The lower response rate for the post-test may be attributed to factors such as competing academic priorities and time constraints, or student engagement around end of the trimester assessment period.

## Study sample

Among the 80 students 86% identified as female, 13% as male and 1% (n = 1) preferred not to disclose their gender. This person was removed from analysis and 79 students were involved in the analysis. The age distribution showed that 31% of students were

**Table 3. Descriptors and associated score values for Total Confidence and subscales within the IPE ABC Questionnaire.**

| | Scale Scores | Very Confident Range | Moderately confident range | Slightly confident range | Not confident at all Range |
|---|---|---|---|---|---|
| Total Confidence | 0-152 | 114-152 | 77-113 | 39-76 | 0-38 |
| Interprofessional Teamwork | 0-80 | 61-80 | 41-60 | 21-40 | 0-20 |
| Behaviours Underpinning Collaboration/Interprofessional Communication | 0-36 | 28-36 | 19-27 | 10-18 | 0-9 |

in the 18–21 age range, while 69% were over 21. The sample included students from seven different professional groups. Nursing students, regardless of their branch, comprised 69% of the sample (n=55), while allied health professionals accounted for 31% (n=25). The variation of sample sizes across professional groups reflects the distribution of student enrolment within the study population. Nursing students were more predominantly represented due to their large overall numbers in the educational institution.

### Pre-test scores

**Total confidence (TC).** The overall mean for the total sample before the module was 108.52 (95% Confidence Interval [CI] 104.43–112.61) indicating a moderate level of self-perceived confidence. The moderate confidence levels were consistent across age, gender, or professional group, with no significant differences identified in pre-test scores based on these variables. (Tables 4 and 5).

**Subscale scores: ITW, BUC and IC.** The results indicated moderate confidence levels in the sub scales of ITW and IC; with mean scores of 57.11 (95% CI 54.91–59.30) and 23.01 (95% CI 21.77–24.26) respectively. This suggests that pre the module students demonstrated a moderate level of self-perceived confidence. This moderate level of confidence was consistent across age, gender, or professional groups, with no significant differences in pre-test scores based on these variables (Tables 4 and 5).

For the BUC subscale, students reported very confident levels with a mean score of 28.40 (95% CI 27.28–29.51.) Four professional groups reported as being very confident as did female students and students within the age range of 18–40. Despite this trend no significant differences were identified in pre-test scores based on these variables (Tables 4 and 5).

### Pre and post-test Scores

Table 6 presents the total sample pre and post intervention scores for self-perceived confidence for the IPE-ABC scale total confidence and subscales. All post test scores increased post the intervention.

### Total confidence

Post intervention, TC increased across all groups to very confident levels of self-perceived confidence. Students under 40 demonstrated higher levels of self-perceived confidence (18–21; M=132.32, 22–30: M=126.39, 31–40: M=129.40) compared to older students. Male students appeared more confident (M=129.25)) than females, and Podiatry students demonstrated the highest TC score (M=135.77).

Pre-test and post-test scores are presented in Table 6. Pre-test self-perceived scores for TC (M=108.25, SE 1.10) increased post module completion (M=128.06, SE 1.88). The difference −19.54, (95% CI) [−24.47–14.61] was significant (t (78) = −7.89, p =<.001), representing a large size effect (d =.888).

### Subscale results: ITW, BUC and IC

Post the intervention ITW increased to very confident levels across all groups. Students younger than 40 demonstrated higher levels of self-perceived confidence (18–21: M=69.50, 22–30: M=66.75, 31–40: M=68.50). Male students were more confident (M=68.20) than females, with Podiatry students showing the highest ITW score (M=71.83).

**Table 4. Mean results for pre-test and post-test for Total Confidence and subscales within the IPE-ABC Questionnaire for Age and Gender.**

| Age and Gender | n | Total Confidence Pre module M (SD) Post module M (SD) | Interprofessional Team Working Pre module M (SD) Post module M (SD) | Behaviours Underpinning Collaboration Pre module M (SD) Post module M (SD) | Interprofessional Communication Pre module M (SD) Post module M (SD) | Statistical Results |
|---|---|---|---|---|---|---|
| 18-21 | 25 | 110.77 (15.38) **132.32 (13.44)** | 57.96 (8.61) **69.50 (7.48)** | **29.49 (4.94)** **33.25 (2.93)** | 23.32 (4.04) **29.57 (4,28)** | **Age (Pre-test)** Total Confidence: F=1.530 df=3,106, *p=0.214*, $\eta^2=0.058$ Interprofessional teamwork: F=0.760, df=3,106, *p=0.520*, $\eta^2=0.30$ Behaviours underpinning collaboration: F=1.536, df=3, 106, *p=0.212*, $\eta^2=0.058$ Interprofessional communication: F=2.527, df=3,106, *p=0.064*, $\eta^2=0.092$. Age (Post Test) Total Confidence: F=1.52, df=3,106, p=0.215, $\eta^2=0.57$ Interprofessional teamwork: F=1.487, df=3,106, *p=0.225* $\eta^2=0.056$ Behaviours Underpinning collaboration: F=1.028, df=3,106, *p=0.385* $\eta^2=0.040$ Interprofessional communication: F=1.034, df=3,106, *p=0.282* $\eta^2=0.040$ |
| 22-30 | 24 | 107.48 (18.36) **126.39 (19.55)** | 57.15 (10.25) **66.75 (10.23)** | **28.04 (4.68)** **31.42 (6.88)** | 22.29 (5.89) **28.22 (5.63)** | |
| 31-40 | 20 | 112.42 (21.06) **129.40 (17.23)** | 58.25 (10.46) **68.50 (9.00)** | **28.94 (5.04)** **31.94 (4.29)** | 25.23 (6.69) **28.95 (5.30)** | |
| 41 and over | 11 | 98.77 (19.00) **119.99 (13,58)** | 53.09 (11.12) **63.00 (8.07)** | **25.80 (5.37)** **30.44 (3.53)** | 19.88 (4.91) 26.55 (3.47) | |
| Female | 69 | 109.25 (17.48) **127.89 (16.61)** | 57.43 (9.38) **67.41 (8.97)** | **28.63 (4.87)** **31.85 (5.05)** | 23.19 (5.53) **28.63 (4.59)** | **Gender (Pre-test)** Total Confidence: t=0.939, df=77, *p=0.175*, d=0.318 Interprofessional teamwork: t=0.761, df=77, *p=0.225*, d=0.257 Behaviours underpinning collaboration=1.099, df, 77, *p=0.138*, d=0.372 Interprofessional communication: t=0.757, df=77. **p=0.226*, d=0.256 *Gender (Post-Test)* *Total Confidence:* t=− 0.239, df=77, *p=0.406*, d=−0.081 Interprofessional teamwork: t=−0.259, df=77, *p=0.398*, d=−0.088 Behaviours underpinning collaboration=−0.565, df, 77, *p=0.223*, d=−0.191 Interprofessional communication: t=0.217, df=77. **p=0.414*, d=0.074 |
| Male | 10 | 103.45 (23.35) **129.25 (18.51)** | 54.90 (12.67) **68.20 (9.47)** | 26.78(5.66) **32.78 (3.24)** | 21.76 (5.89) **28.26 (7.07)** | |
| Total matched pairs | 80 | 108.52 (18.25) **128.06 (16.74)** | 57.11 (9.80) **67.51 (8.97)** | **28.40 (4.98)** **31.97 (4.85)** | 23.01 (5.56) **28.58 (4.91)** | |

M=Mean, SD=Standard Deviation, df=degrees freedom an independent samples *t-test or ANOVA (with a Levene's test indicating homogeneity of variance across groups) indicated no statistical differences in groups based on gender or age in the pre –test scores.

**Underlined** means indicate very confidence levels of confidence. Cohen's d (d) and eta squared ($\eta^2$) produced as measures of effect size. **NB: one student did not define their gender and thus could not be included in analysis.**

Pre-test ITW scores (M=57.10, SE 1.10) increased post-module (M=67.50, SE1.01.). This difference −10.40, 95%CI [−13.07, −7.73] was significant t (78) −7.47, p =<.001 and represented a large size effect d = .872.

Similar patterns were seen in the BUC subscale which increased in all groups to very confident levels post intervention. Students younger than 40 reporting higher levels of confidence (18–21; M=33.25, 22–30: M=31.42, 31–40: M=31.94) than older students. Male students appear more confident (M=32.78) than females and Podiatry students demonstrated the highest BUC score (M=34.67).

Pre-test scores for the BUC subscale (M=28.58, Mdn=29.00) increased post module (M=31.97, Mdn 34.00), with significant difference (z=4.750, p=<.001), representing a medium effect size (r = .534).

**Table 5. Mean results for pre-test and post-test Total Confidence and subscales within the IPE-ABC Questionnaire for professional group.**

| | n | Total Confidence Pre module M (SD) Post module M (SD) | Interprofessional Team Working Pre module M (SD) Post module M (SD) | Behaviours Underpinning Collaboration Pre module M (SD) Post module M (SD) | Interprofessional Communication Pre module M (SD) Post module M (SD) | Statistical Results |
|---|---|---|---|---|---|---|
| Adult Nursing | 22 | 109.52 (15.99) __130.39 (15.82)__ | 57.89 (7.95) __68.59 (9.00)__ | __28.27 (4.45)__ __31.95 (3.91)__ | 23.36 (5.94) __29.85 (4.67)__ | **Professional Group (Pre-test)** Total Confidence: F= .161, df =7,73, *p=0.986,* η² = 0.013 |
| Child Nursing | 6 | 110.17 (10.05) __121.83 (30.47)__ | 59.00 (4.98) __64.00 (13.80)__ | __30.33 (3.67)__ __28.67 (12.23)__ | 20.83 (4.22) __29.17 (5.85)__ | Interprofessional teamwork: F=.376, df=7,73, *p=0.892* η² = 0.30 |
| Diagnostic Imaging Radio-therapy and Oncology | 12 | 109.15 (23.51) __128.02 (14.30)__ | 57.36 (12.56) __67.87 (8.48)__ | __28.91 (5.27)__ __32.09 (3.90)__ | 22.88 (6.89) 28.06 (3.75) | Behaviours underpinning collaboration: F =.220, df=6, *p=0.969* η² = 0.018. Interprofessional communication: F= .262, df, 7,73, *p=0.953,* η² =0.021 |
| Learning Dis-ability Nursing | 12 | 111.26 (16.06) __128.42 (16.56)__ | 59.08 (8.77) __67.83 (8.92)__ | 27.99 (5.72) __31.75 (4.29)__ | 24.19 (5.39) __28.83 (4.80)__ | Professional Group (Post Test) Total Confidence: F = 0.514, df = 7,73, p = 0.796, η² = 0.041 |
| Mental Health Nursing | 15 | 105.20 (21.45) __124.99 (14.59)__ | 54.47 (11.39) __66.07 (8.33)__ | 27.80 (5.60) __32.32 (3.65)__ | 22.93 (5.81) 26.60 (4.87) | Interprofessional teamwork: F = 0.556, df = 7,73, *p = 0.764* η² = 0.044 |
| Occupational Therapy | 7 | 106.12 (18.25) __125.51 (14.49)__ | 54.71 (9.41) __65.36 (8.22)__ | __28.69 (5.76)__ __32.00 (3.83)__ | 22.71 (4.07) __28.14 (3.48)__ | Behaviours underpinning collaboration: F = 0.779, df = 7, 73, *p = 0.589,* η² = 0.061 |
| Podiatry | 6 | 107.64 (26.28) __135.77 (17.62)__ | 57.33 (15.03) __71.83 (7.88)__ | 27.97 (5.61) __34.67 (2.16)__ | 22.33 (6.47) __29.27 (8.27)__ | Interprofessional communication: F = 0.703, df = 7,73, *p = 0.648* η² = 0.055 |

M = Mean, SD = Standard Deviation, df = degrees freedom an independent samples *t-test or ANOVA (with a Levene's test indicating homogeneity of variance across groups) indicated no statistical differences in groups based on profession in the pre–test scores. **Underlined** means indicate very confidence levels of confidence. Cohen's d (d) and eta squared (η²) produced as measures of effect size.

IC scores increased post intervention across the board, with older students (41+) showing moderate levels confidence (M = 26.55), as did Mental Health Nursing students (M = 26.60). Adult Nursing students had the highest level of confidence in this subscale (M = 29.85). Pre-test scores (M = 23.11, SE.63) increased post module (M = 28.58, SE.55), p =<.001 and represented a large size effect d = .945.

## Between subject effects

**Total confidence.** There was a significant effect of time on self-reported levels of confidence, F (1) =24.62, **p < 0.001,** suggesting a positive impact of the module on student confidence levels. There were no significant interactions between time and gender (F (1) =.275, p0.602). or time and age (F (3) =.298, p0.827) or time and profession (F (6) =.238, P0.962). This indicates that time was the main influencing factor.

**Interprofessional teamworking.** There was a significant effect of time on ITW confidence, F (1) =22.26, **p < 0.001**, suggesting a positive impact of the module on student confidence levels. There were no significant interactions between time and gender (F (1) =.14.19, p0.672), or time and age (F (3) =.181, p0.909), or time and profession (F (6) =.280, p0.945). Similarly, this indicates that time was the main influencing factor n students reported level of confidence.

**Behaviours underpinning collaboration.** There was a significant effect of time on BUC confidence, F (1) =8.88, **p 0.004**, suggesting a positive impact of the module. There was no significant interaction between time and gender (F (1) =.345, p0.559), or time and age (F (3) =.272, p0.845), or time and profession (F (6) =.702, p0.649). This indicates that time was the main influencing factor on students reported level of confidence.

**Interprofessional communication.** There was a significant effect of time on IC confidence, F (1) = 32.58, **p < 0.001**, suggesting a positive impact of the module on student confidence levels. There was no interaction between time and

**Table 6. Differences between the pre-test and post-test IPE ABC scores for all students.**

| Scale and subscale | Pre-test Mean M (SD) Median | Post-test Mean M (SD) Median | t-test (df), p value; Cohen's d, D Willcoxon signed rank z score, p value and r |
|---|---|---|---|
| Total Confidence | 108.52 (18.25) 110.00 | 128.06 (16.74) 128.00 | −7.89 (78) p<.001 d=.888 |
| Interprofessional Teamwork | 57.11 (9.80) 59.00 | 67.51 (8.97) 66.00 | −7.75 (78) p<.001 d=872 |
| Behaviours Underpinning collaboration | 28.40 (4.98) 29.00 | 31.97 (4.85) 34.00 | Z=4.750 p<.001 r=.534 |
| Interprofessional Communication | 23.01 (5.56) 23.00 | 28.58 (4.91) 28.00 | −8.40 (78) p<.001 d=.945 |

M= Mean, SD= Standard Deviation, df = degrees freedom, r = Pearson's r.

gender (F (1) =.188, p0.666), or time and age (F (3) =.837, p0.478), or time and profession (F (6) =.625, p0.709). This indicates that time was the main influencing factor on students reported level of confidence.

## Discussion

We have reported the evaluation of student confidence pre and post completion of a mandatory six-week academic IPE module. Post module completion TC and subscale scores of IPW and BUC increased to very confident levels across all groups. Younger students (< 40), male students and podiatry students demonstrated the highest levels of post intervention confidence. All groups showed improvements, with most substantial gains observed in IC. Between subject analysis demonstrated that time was the only factor impacting on self-perceived levels of confidence for TC and for the subscales of the IPE ABC scale. These results indicate that the module was associated with increased confidence, on students self-perceived levels of confidence. These increases in confidence post intervention are similar to other studies that have measured confidence pre and post an IPE intervention in the academic environment [22,19,35].

Hermansen-Kobulnicky, identified confidence increases after an introductory IPE intervention [22]. Similar to the current study, students developed a patient centred plan based on a paper-based case study using a structured team meeting approach. However, Hermansen-Kobulnicky's work showed significant differences in confidence levels between professional groups when considering the mean pre and post levels of confidence. This is different to the findings in the current study where there were differences in the levels of self-perceived confidence, however the differences were not found to be statistically significant. This difference could be attributed to the smaller sample size and varied group sizes, which may have introduced the possibility of a Type 2 error within the current study. Any future studies should consider larger, balanced samples across professional groups to further explore the relationship.

Like the IPE ABC scale, the constructs used in the Hermansen-Kobulnicky questionnaire [22] were based on Interprofessional Education Competencies ([IPEC], IPEC, 2016) relating to IPE and IPC; including values and ethics for interprofessional practice, roles, and responsibilities, interprofessional collaboration and teams and teamwork. However, the IPE ABC scale also included other known IPE competency frameworks in its development such as the Canadian Interprofessional Health Collaborative competencies [36] and established learning outcomes for Interprofessional Education [37]. Moreover, the IPE ABC scale also underwent extensive psychometric testing to ensure its validity and reliability adding strength to the current study's findings [25].

Attitudes and readiness to engage in learning have been linked to presage factors such as self-efficacy, motivation, gender, age and expectations of learning [25,38–41]. These elements are in turn, linked to academic behavioural confidence [42]). Studies on general academic behavioural confidence have reported gender-based differences, with male students typically exhibiting higher levels of confidence in the academic setting [43]. Similarly, Williams et al. found that male students demonstrated higher levels of self-efficacy, which is a component of academic behavioural confidence when compared to female students within interprofessional education activities, [21].

These results are aligned with the current study in that male students demonstrated non-significantly higher levels of confidence pre and post intervention. Sander and Sanders suggest that male students may be less sensitive to social interaction issues, which could explain the observed differences, as the constructs that underpin the IPE ABC scale are reliant on social interactions [42]. Given that only 10 males' students were included, further investigations with a more balanced gender distribution is recommended as these subgroup comparisons may be underpowered. The lack of statistical significance should therefore be interpreted with caution as it may reflect a Type 2 error rather than a true absence of gender-based differences.

Students in the current study demonstrated lower levels of self-perceived confidence before the UTPL intervention in scores for TC, BUC, and TW (moderate levels) than those demonstrated (very confident) in the original IPE ABC scale validation study. The original validation study involved first year health and social care students, and the current study focussed on third year students. Therefore, it is reasonable to suggest that confidence may reduce in students as they progress through their studies.

Within the IPE literature students tend to assess themselves positively in relation to their attitudes towards IPE and their ability to undertake IPE interactions on commencement of training, however, these positive perceptions in ability and attitude appear to decrease over time [21,44]. This may explain the differences observed in the current study and the original IPE-ABC validation study. The current results suggest that as students' progress through more complex and challenging aspects of the curriculum with increased exposure to interprofessional working, their confidence may be affected. To confirm this, further research should explore longitudinal patterns of student confidence through their education and practice. As part of our commitment to further exploring these findings, the research team has initiated a longitudinal study to assess the sustained impact of IPE interventions over time and across different professional groups.

Over-confidence bias was noted in the IPE ABC validation study [45]. Before the intervention in the current study overall students rated themselves as very confident in behaviours underpinning collaboration, such as respecting, listening to, valuing others, and recognising equality among team members. These are expected behaviours in interprofessional collaborative practice [28] and students may feel pressured to present themselves as confident in these areas, even if they are not. This raises the possibility of measurement error due to social desirability response bias [SDRB] (Van De Mortel [46]). SDRB can occur for a variety of reasons, the individual may self-deceive or present a positive answer to conform to socially acceptable values, avoid criticism or gain social approval [46]. SDRB is most likely to occur in response to socially sensitive questions and as some questions within the IPE-ABC scale may be classed as sensitive it is possible that students may feel they cannot disclose a lack of confidence in certain behaviours. This positive skew in confidence reporting was also observed in the Hermansen-Kobulnicky study, which used a 10-point scale to allow for more nuanced responses and to increase sensitivity in detecting changes [22]. However, it seems that overconfidence bias persists, regardless of the scale used.

In this study, while using a valid and reliable tool such as the IPE-ABC scale helps mitigate social desirability bias to some extent, future research could benefit form including a tool specifically designed to measure SDRB. Further qualitative exploration could provide insight into how students perceive and report their confidence, enabling educators to address potential biases in self-evaluation. To address potential overconfidence and social desirability bias, faculty could consider embedding peer–assessment opportunities, structured reflection and formative feedback within the modules. In

addition, appropriate debriefs within simulated conditions can support and develop students' awareness. These strategies may help students to recognise both strengths and areas of development in collaborative behaviours.

While confidence building is key goal of IPE, equal emphasis must be placed on continuous confidence calibration. This could be achieved through structured reflection and formative feedback mechanisms to ensure students develop sufficient self-awareness and self-assessment abilities. Additionally, this could be achieved through using skilled faculty who should create a psychologically safe space for students to learn and reflect [47]. For creating psychologically safe environments, educators may consider implementing structured peer assessment, guided self-evaluation checklists, and facilitated debriefing sessions to help students calibrate their confidence and develop successful self-assessment skills.

While the results indicate increased confidence within an academic module, this does not directly demonstrate an effect on real-world interprofessional practice. Future research should explore whether these gains in confidence continue and positively influence actual collaborative behaviours in clinical or placements settings.

## Limitations

Limitations of the current study relate to the differences in student population based on professional groups, age, and gender. We acknowledge the absence of a priori power analysis to justify the sample size as it was determined on feasibility and convenience. Groups sizes were unequal, and sub-analysis involved small samples increasing the possibility of a Type 2 error and studies with larger sample sizes are needed to mitigate this risk. The study response rate was low with only 80 matched pairs being evaluated pre and post the UTPL intervention. This could be due to participant error in alias code completion which could have impacted negatively on ability to match participants in both stages of the study. The reduced response rate introduces the possibility of non-response bias as the final sample may not fully present the entire student cohort. It is plausible only the students with positive perceptions of IPE completed the questionnaire and which may explain confidence measures increased. Additionally, the low response rate may introduce nonresponse bias, as those who choose to participate may have held more positive perceptions of IPE than non-responders. Moreover, as the study measured confidence within an academic module only, the findings may not be directly generalisable to real-world clinical practice. The sampling frame used in the current study is limited by the students that undertook the UTPL module in trimester B and it is unknown if the results would be replicated in other professionals who undertook the module in trimester A (Social Work, Physiotherapy, Human Nutrition and Dietetics, Oral Health and Orthoptic students).

Additionally, gender was explored as a potential variable with 86% of the total 80 students identified as female. The original IE ABC had identified gender played a role with male students more confident than female students. However, the small sample sizes and unequal gender distribution limits drawing any definitive conclusions about gender differences in this study. Furthermore, the study only measured self-perceived confidence in the academic setting, leaving it unknown whether this confidence translates to the practice settings into collaborative practice. To address this gap further research is being planned to explore the impact of academic interprofessional education on interprofessional collaboration in new graduates.

While the results suggest that the UTPL model improved student confidence after a six-week academic intervention, Hermansen-Kobulnicky [22] undertook a case based embedded assignment intervention of an unspecified duration and demonstrated similar results. This raises questions about the duration and type of IPE needed to affect confidence, and if it is reasonable to consider that shorter IPE events may produce similar results [22]. Future studies could explore the minimum duration required to achieve significant improvements in student confidence.

Regarding the use of the 4-point Likert scale, students, as in other confidence-related research, tended to rate themselves on the higher end of the scale. This suggests a potential ceiling effect and limits inferences that can be drawn between individuals scoring at the higher end of the IPE-ABC scale and may suggest an issue with the sensitivity of the tool. These findings could also reflect SDRB where students wish to portray themselves in a positive light in relation to being able to undertake behaviours underpinning teamwork and collaboration, introducing the possibility of measurement

errors. This could reflect SDRB where students wish to portray themselves in a positive light in relation to being able to undertake behaviours underpinning teamwork and collaboration, introducing the possibility of measurement errors. Despite these limitations, the use of the validated IPE ABC scale strengthens the reliability of this study, supporting the need for structured, evidence-based IPE interventions to prepare students for interprofessional practice effectively.

Finally, the study acknowledges potential confounding factors that may have influenced the results. Specifically, variations in students prior experience, motivation levels, and the impact of time on confidence development were not fully controlled for. Prior experience in IPE settings may lead to differing changes to baseline confidence scores. The authors also acknowledge that the study was conducted in an academic environment and therefore the transferability of the results to the practice setting is unknown. In addition, the study did not include a control group, limiting our ability to draw causal conclusions about the effect of the intervention.

## Conclusion

This study used a pre-post-test design to assess student confidence of undergraduate pre-registration health and social care in relation to IPE related behaviours using the validated IPE-ABC scale. Students participated in a six-week interprofessional module using a TOSPE approach. Before the module students demonstrated moderate levels of confidence for TC and for the subscales of ITW and IPC. For the behaviours underpinning collaboration, students perceived themselves to be very confident. Following the intervention, student confidence increased to very confident across all subscales of the IPE ABC scale, and these changes were statistically significant. When considering between subject effects, only time was found to demonstrate significant results indicating an association between the IPE intervention and increased self-perceived confidence. It should be noted that factors such as social desirability response bias, non response bias, over-confidence bias and Type 2 errors relating to small subgroup sample sizes may impact the results observed and hence interpretation of the study findings should consider these limitations.

This study contributes to the existing literature by reinforcing the positive influence of IPE interventions on self-perceived behaviours that support collaborative practice. However, whether these self-reported improvements translate into actual collaborative behaviours in clinical practice remains to be confirmed. Validated self-report tools such as the IPE-ABC can support curriculum developers in monitoring and adjusting pre-registration IPE design.

## Supporting information

**S1 Text. Data File Exploring Confidence Development in Interprofessional Teams.**
(XLSX)

**S2. STROBE checklist cohort.**
(DOCX)

## Acknowledgments

The authors would like to acknowledge students and staff who participated and supported this research.

## Author contributions

**Conceptualization:** Sharron Blumenthal, Jamie McDermott, John Locke, Lindsey Burns.

**Data curation:** Sharron Blumenthal, Lindsey Burns.

**Formal analysis:** Sharron Blumenthal, Kareena McAloney-Kocaman, Lindsey Burns.

**Investigation:** Sharron Blumenthal, Lindsey Burns.

**Methodology:** Sharron Blumenthal, Jamie McDermott, Kareena McAloney-Kocaman, Lindsey Burns.

**Project administration:** Sharron Blumenthal, Lindsey Burns.

**Validation:** Sharron Blumenthal, Lindsey Burns.

**Writing – original draft:** Sharron Blumenthal, Sivaramkumar Shanmugam, Jamie McDermott, John Locke, Tamsin Fitzgerald, Christopher Duncan, Kareena McAloney-Kocaman, Lindsey Burns.

**Writing – review & editing:** Sharron Blumenthal, Sivaramkumar Shanmugam, Jamie McDermott, John Locke, Tamsin Fitzgerald, Christopher Duncan, Kareena McAloney-Kocaman, Lindsey Burns.

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
