## [Decision Letter · Decision Letter 0]

PONE-D-24-55482Exploring Confidence Development in Interprofessional Teams: A Pre-Post Analysis of a Health and Social Care Education ModulePLOS ONE

Dear Dr. Shanmugam,

Thank you for submitting your manuscript to PLOS ONE. After careful consideration, we feel that it has merit but does not fully meet PLOS ONE’s publication criteria as it currently stands. Therefore, we invite you to submit a revised version of the manuscript that addresses the points raised during the review process.

We look forward to receiving your revised manuscript.

Kind regards,

Javier Fagundo-Rivera, PhD

Academic Editor

PLOS ONE

Journal Requirements:

3.  Please include a separate caption for figure 1 in your manuscript.

5. We note that there is identifying data in the Supporting Information file < Data File for PLOSOne Submission.xls>. Due to the inclusion of these potentially identifying data, we have removed this file from your file inventory. Prior to sharing human research participant data, authors should consult with an ethics committee to ensure data are shared in accordance with participant consent and all applicable local laws.

-Location data

Please remove or anonymize all personal information, ensure that the data shared are in accordance with participant consent, and re-upload a fully anonymized data set. Please note that spreadsheet columns with personal information must be removed and not hidden as all hidden columns will appear in the published file.

6. We note that this data set consists of interview transcripts. Can you please confirm that all participants gave consent for interview transcript to be published?

If they DID provide consent for these transcripts to be published, please also confirm that the transcripts do not contain any potentially identifying information (or let us know if the participants consented to having their personal details published and made publicly available). We consider the following details to be identifying information:

- Names, nicknames, and initials

- Age more specific than round numbers

- GPS coordinates, physical addresses, IP addresses, email addresses

- Information in small sample sizes (e.g. 40 students from X class in X year at X university)

- Specific dates (e.g. visit dates, interview dates)

- ID numbers

Or, if the participants DID NOT provide consent for these transcripts to be published:

- Provide a de-identified version of the data or excerpts of interview responses

- Provide information regarding how these transcripts can be accessed by researchers who meet the criteria for access to confidential data, including:

a) the grounds for restriction

b) the name of the ethics committee, Institutional Review Board, or third-party organization that is imposing sharing restrictions on the data

c) a non-author, institutional point of contact that is able to field data access queries, in the interest of maintaining long-term data accessibility.

d) Any relevant data set names, URLs, DOIs, etc. that an independent researcher would need in order to request your minimal data set.

For further information on sharing data that contains sensitive participant information, please see: https://journals.plos.org/plosone/s/data-availability#loc-human-research-participant-data-and-other-sensitive-data

If there are ethical, legal, or third-party restrictions upon your dataset, you must provide all of the following details (https://journals.plos.org/plosone/s/data-availability#loc-acceptable-data-access-restrictions):

1. A complete description of the dataset

2. The nature of the restrictions upon the data (ethical, legal, or owned by a third party) and the reasoning behind them

3. The full name of the body imposing the restrictions upon your dataset (ethics committee, institution, data access committee, etc)

4. If the data are owned by a third party, confirmation of whether the authors received any special privileges in accessing the data that other researchers would not have

5. Direct, non-author contact information (preferably email) for the body imposing the restrictions upon the data, to which data access requests can be sent

**Additional Editor Comments:**

Dear authors,

Four reviewers have reviewed your manuscript and a Major Revision is recommended.

Please, respond to all these comments in your Response Letter.

Also, consider that Reviewer 2 comments have been attached separately.

We look forward to your responses.

Best regards.

Reviewers' comments:

Reviewer's Responses to Questions

**Comments to the Author**

1. Is the manuscript technically sound, and do the data support the conclusions?

Reviewer #1: Yes

Reviewer #2: Yes

Reviewer #3: Yes

Reviewer #4: Yes

2. Has the statistical analysis been performed appropriately and rigorously? 

Reviewer #1: Yes

Reviewer #2: Yes

Reviewer #3: Yes

Reviewer #4: Yes

3. Have the authors made all data underlying the findings in their manuscript fully available?

Reviewer #1: Yes

Reviewer #2: Yes

Reviewer #3: Yes

Reviewer #4: Yes

4. Is the manuscript presented in an intelligible fashion and written in standard English?

Reviewer #1: Yes

Reviewer #2: Yes

Reviewer #3: Yes

Reviewer #4: Yes

5. Review Comments to the Author

**Reviewer #1: **

1- The “Introduction” should be modified, rearranged and divided into some paragraphs based on scientific writing. Paragraph 1 needs to incorporate brief yet comprehensive introduction of Confidence Development in Inter professional Teams and the main question of the research work.

2- The author(s) should modify relevant key words based on MESH.

The “Methodology” should be modified, rearranged and divided into some paragraphs based on scientific writing. this section should mention the phases of study very clearly and justify the usage of any statistical tools and approaches one by one in this regard.

3- It is not mentioned how and by which statistical test the validity and reliability of questionnaire were examined.

4- The authors should mention and justify why The sample sizes across different professional groups are unequal, which could reduce the statistical power of subgroup analyses,

5- The authors should justify the low response rate (26% for post-test) that introduces nonresponse bias, meaning the results may not be representative of the entire student cohort.

6- In samples Among the 80 students 86% identified as female, Dose Gender have any influence on the result of the research? please mention it.

7- The authors should mention how they control the confounding factors in this research like time, prior experience, different motivation levels of student, etc.

8. The "Result", "discussion" and "Conclusion" should be written based on scientific writing approach and also arranged based on the main and specific objectives of the research work.

9. I found it hard to follow the paper, Consistency and coherency of whole manuscript is weak and should be improved.

**Reviewer #2: **

Comments to the authors are provided in the attached document. The manuscript presents a pre-post study examining the impact of a 6-week academic interprofessional education (IPE) module on students’ self-perceived confidence in interprofessional teamwork.

See document attached.

**Reviewer #3: **

1. Abstract

Strengths:

Clearly presents the aim, methods, results, and conclusions.

Inclusion of effect sizes and significance values is commendable.

Recommendations:

Avoid causative language like “suggesting the IPE intervention had a positive impact” unless supported by stronger experimental controls.

Suggested revision: “indicating an association between the IPE intervention and increased self-perceived confidence.”

2. Introduction

Strengths:

Comprehensive overview of IPC and IPE significance, backed by strong references.

Suggestions:

The introduction would benefit from sharper articulation of the knowledge gap. It currently blends motivation with background without clearly stating what this study adds.

Suggested addition: “Despite widespread endorsement of IPE, few studies have evaluated validated tools in pre-registration academic settings using robust pre-post designs.”

3. Background

Strengths:

Theoretical framework (self-efficacy, expectancy-value theory) is well-integrated.

Suggestions:

The background could be more concise. Some of the historical institutional information may be more appropriate for supplementary material unless critical for methodology.

4. Methods

Strengths:

Methodology is described in strong detail with ethical approvals, sampling, instruments, and analysis procedures clearly stated.

Use of a validated scale (IPE-ABC) and description of subscales adds robustness.

Recommendations:

Sampling bias is likely due to low response rates (26% post-test). This should be acknowledged more prominently in the methods or limitations.

Data matching by alias codes is creative, but the risk of participant error or dropouts affecting match quality should be noted.

5. Results

Strengths:

Clear presentation of results across subscales with robust statistical analysis (paired t-tests, Wilcoxon tests, ANOVA).

Use of both Cohen’s d and Pearson’s r for effect size is a good practice.

Suggestions:

Use of negative values for Cohen’s d (e.g., d = –0.888) is unconventional. Since effect size direction is not meaningful in pre-post designs where direction is known, report absolute values for clarity.

Discuss implications of high baseline scores and possible ceiling effects—several participants were already “very confident” pre-module.

Tables are comprehensive but visual aids (e.g., bar charts or boxplots) could help illustrate change distributions.

6. Discussion

Strengths:

Strong integration with previous studies

Addresses psychological theories (self-efficacy, overconfidence bias, SDRB), which adds analytical depth.

Suggestions:

The narrative is overly descriptive in places; streamline and focus on key implications (e.g., the need for continuous confidence calibration, not just confidence building).

Provide practical recommendations for educators: how might one modify IPE modules to address overconfidence or SDRB?

Highlight potential Type II error risk more explicitly in subgroup comparisons (e.g., small male sample).

7. Limitations

Strengths:

Thoughtfully acknowledges low response rate, sample imbalance, and SDRB.

Plans for longitudinal follow-up are commendable.

Additional Suggestions:

Add a statement on generalizability—results may not apply to students in non-academic or clinical IPE settings.

Clarify whether the low male participation is reflective of program enrollment or introduces potential gender bias in outcomes.

8. Conclusion

Strengths:

Accurately reflects the findings without overstating them.

Reinforces the contribution of validated tools to IPE evaluation.

Recommendation:

Emphasize the need for future research on the translation of academic confidence into clinical settings, especially during transition to practice.

9. Technical Aspects & PLOS ONE Fit

Fit for Journal:

The topic aligns well with PLOS ONE’s focus on interdisciplinary, educational, and public health research.

Writing & Structure:

Language is professional and generally clear, though some sections (e.g., Background, Discussion) are verbose and would benefit from editing for brevity.

**Reviewer #4: **

I found the paper very interesting and well written and organized. I would like some more graphics but the information was still all there. Just a few small suggestions - to include a description of the acronym IPE in the abstract and I wasn't sure I completely understood the key of table 4, in particular: very confidence levels of confidence sounded odd. Also in the results for example it was stated that the confidence level in males was higher but only provided the value of confidence for males.

But they are very minor comments. Overall I found the paper very interesting and well written.

6. PLOS authors have the option to publish the peer review history of their article (what does this mean? ). If published, this will include your full peer review and any attached files.

**Do you want your identity to be public for this peer review?** For information about this choice, including consent withdrawal, please see our Privacy Policy .

Reviewer #1: No

Reviewer #2: No

Reviewer #3: **Yes: ** Juan Carlos Alvarado Gonzalez. MD

Reviewer #4: No

---

## [Author Response · Author response to Decision Letter 1]

21 May 2025

Please refer to the uploaded "Response to Reviewers" document, which outlines all the actions taken and rebuttals based on the feedback and comments provided.

---

## [Decision Letter · Decision Letter 1]

PONE-D-24-55482R1Exploring Confidence Development in Interprofessional Teams: A Pre-Post Analysis of a Health and Social Care Education ModulePLOS ONE

Dear Dr. Shanmugam,

Thank you for submitting your manuscript to PLOS ONE. After careful consideration, we feel that it has merit but does not fully meet PLOS ONE’s publication criteria as it currently stands. Therefore, we invite you to submit a revised version of the manuscript that addresses the points raised during the review process.

We look forward to receiving your revised manuscript.

Kind regards,

Javier Fagundo-Rivera, PhD

Academic Editor

PLOS ONE

Journal Requirements:

**Additional Editor Comments:**

Dear Authors,

I am pleased to inform you that two reviewers have recommended the publication of your manuscript, while the remaining two reviewers have suggested minor revisions (see attached document for Reviewer 2). Accordingly, we invite you to submit a revised version of your paper addressing the reviewers’ comments.

Please revise your manuscript to incorporate the requested changes and provide a point-by-point response to each reviewer’s remarks.

Thank you for your contribution, and we look forward to receiving your revised manuscript.

Sincerely,

Reviewers' comments:

Reviewer's Responses to Questions

**Comments to the Author**

1. If the authors have adequately addressed your comments raised in a previous round of review and you feel that this manuscript is now acceptable for publication, you may indicate that here to bypass the “Comments to the Author” section, enter your conflict of interest statement in the “Confidential to Editor” section, and submit your "Accept" recommendation.

Reviewer #1: All comments have been addressed

Reviewer #2: All comments have been addressed

Reviewer #3: All comments have been addressed

2. Is the manuscript technically sound, and do the data support the conclusions?

Reviewer #1: Yes

Reviewer #2: Yes

Reviewer #3: Yes

3. Has the statistical analysis been performed appropriately and rigorously? 

Reviewer #1: Yes

Reviewer #2: Yes

Reviewer #3: Yes

4. Have the authors made all data underlying the findings in their manuscript fully available?

Reviewer #1: Yes

Reviewer #2: No

Reviewer #3: Yes

5. Is the manuscript presented in an intelligible fashion and written in standard English?

Reviewer #1: Yes

Reviewer #2: Yes

Reviewer #3: Yes

6. Review Comments to the Author

Reviewer #1: ACCEPT

Reviewer #2: This manuscript presents an original empirical study that evaluates the impact of an interprofessional education module on the self-perceived confidence of students in health sciences and social services. The focus is relevant and timely, addressing a gap in literature using a validated tool and appropriate statistical analyses. The work is suitable for a journal like PLOS ONE, with some minor revisions that I provided in an attached document.

Reviewer #3: 

1. Response to Previous Comments

Authors provide a point-by-point response that is thorough and constructive.

Many suggestions (e.g., clarification of Likert scaling, sampling strategy, overconfidence bias) are explicitly addressed in both the revised manuscript and the rebuttal.

Areas Still Requiring Attention:

The language around causality in parts of the discussion and abstract could still be further softened to emphasize associative, not causal, findings.

2. Abstract

Revisions Assessed:

Added language to reflect pre-post design limitations.

Numerical clarity (e.g., effect sizes, significance) retained.

Remaining Suggestion:

Consider changing phrasing from “the module had a positive impact” to “the module was associated with increased confidence,” to maintain alignment with observational design.

3. Discussion

Improvements:

Discussion better balances self-efficacy growth with concerns about overconfidence.

Integration of previous studies is more concise and analytically relevant.

Still Needed:

Though improved, some speculative comments on “real-world applicability” (e.g., bridging to practice) could benefit from hedging language due to the study’s academic-only context.

Positive Additions:

Theoretical linkages to Vygotsky’s ZPD and Bandura’s social cognitive theory remain insightful and are now more clearly aligned with observed outcomes.

4. Conclusion

Improved Version:

Revised to reflect associative findings, rather than overstate intervention effects.

Emphasizes the utility of validated tools for evaluating IPE.

Optional Refinement:

Consider ending with a more actionable statement, such as: “Validated self-report tools such as IPE-ABC can support curriculum developers in measuring and adjusting pre-registration IPE design.”

5. Overall Structure and PLOS ONE Fit

Clarity and Organization:

The paper reads clearly and is logically structured. All sections are complete and professional.

Formatting and Ethics:

Ethical statements, data availability, and adherence to reporting standards (e.g., STROBE for observational designs) are well addressed.

Journal Scope Fit:

The manuscript remains a good fit for PLOS ONE under the categories of health education, interdisciplinary training, and public health implementation research.

7. PLOS authors have the option to publish the peer review history of their article (what does this mean? ). If published, this will include your full peer review and any attached files.

**Do you want your identity to be public for this peer review?** For information about this choice, including consent withdrawal, please see our Privacy Policy .

Reviewer #1: No

Reviewer #2: No

Reviewer #3: **Yes: ** Juan Carlos Alvarado Gonzalez MD, MSc.

---

## [Author Response · Author response to Decision Letter 2]

17 Jun 2025

We have addressed all the feedback given to us in the second round of minor revisions.

---

## [Editor Report · Decision Letter 2]

Exploring Confidence Development in Interprofessional Teams: A Pre-Post Analysis of a Health and Social Care Education Module

PONE-D-24-55482R2

Dear Dr. Shanmugam,

We’re pleased to inform you that your manuscript has been judged scientifically suitable for publication and will be formally accepted for publication once it meets all outstanding technical requirements.

Kind regards,

Javier Fagundo-Rivera, PhD

Academic Editor

PLOS ONE

Additional Editor Comments (optional):

The authors have properly responded to the Reviewers and this manuscript can be finally accepted.

---

## [Editor Report · Acceptance letter]

PONE-D-24-55482R2

PLOS ONE

Dear Dr. Shanmugam,

I'm pleased to inform you that your manuscript has been deemed suitable for publication in PLOS ONE. Congratulations! Your manuscript is now being handed over to our production team.

Kind regards,

on behalf of

Dr. Javier Fagundo-Rivera

Academic Editor

PLOS ONE